# Charting cellular differentiation trajectories with Ricci flow

Anthony Baptista [1,2] ✉, Ben D. MacArthur[1,3,4] & Christopher R. S. Banerji [1,5] ✉

Complex biological processes, such as cellular differentiation, require intricate rewiring of intra-cellular signalling networks. Previous characterisations revealed a raised network entropy underlies less differentiated and malignant cell states. A connection between entropy and Ricci curvature led to applications of discrete curvatures to biological networks. However, predicting dynamic biological network rewiring remains an open problem. Here we apply Ricci curvature and Ricci flow to biological network rewiring. By investigating the relationship between network entropy and Forman-Ricci curvature, theoretically and empirically on single-cell RNA-sequencing data, we demonstrate that the two measures do not always positively correlate, as previously suggested, and provide complementary rather than interchangeable information. We next employ Ricci flow to derive network rewiring trajectories from stem cells to differentiated cells, accurately predicting true intermediate time points in gene expression time courses. In summary, we present a differential geometry toolkit for understanding dynamic network rewiring during cellular differentiation and cancer.

Cellular differentiation is a complex biological process essential for embryonic development as well as the maintenance and repair of adult tissues. Aberrant differentiation underlies a wide spectrum of pathology. This includes malignancy, where cells may fail to differentiate or de-differentiate, becoming trapped in a more plastic, proliferative state[1]. A key feature of cellular differentiation is an orchestrated shift in the intra-cellular transcriptomic distribution. C. H. Waddington proposed in 1939, a seminal interpretation of the intra-cellular state during differentiation, known as the Waddington Landscape[2]. Under this landscape, less differentiated cells occupy a higher potential energy, represented by an elevated position. As cells differentiate they roll down this complex landscape, following a trajectory determined by its hills and valleys, dropping in potential energy until the cell arrives at an attractor state: the differentiated cell.

While an intuitive and appealing picture, the deep complexity of the intra-cellular state revealed by modern transcriptomic and proteomic quantification, as well as the discovery that we can reprogramme cells to earlier phases of differentiation, motivated a recasting of the Waddington Landscape from a metaphorical picture into an interpretable mathematical framework[3,4]. Modern interpretations of Waddington's Landscape have re-framed cell fate trajectories via the phase space of transcriptomic dynamics[5–7]. While non-deterministic elements of these transcriptomic dynamics have motivated more information-theoretic characterisations of cell fate trajectories[8]. The latter interpretation has revealed the intra-cellular states of less differentiated cells can be considered more "promiscuous", displaying a higher entropy in their protein-protein interactions, which decreases during differentiation and increases in cancer, providing a quantitative correlate for the "height" in Waddington's landscape[9–11].

As Waddington's landscape has evolved from an intuitive picture to a mathematical framework, however, cell fate transitions have maintained a geometric appeal[12]. Geometric approaches to studying cell fate have often focused on characterisations of the underlying dynamical system and typically require detailed knowledge of gene-regulatory networks relevant to specific cell fate transitions[7,13]. However, at the genome-wide scale, we do not have this deep

[1]The Alan Turing Institute, The British Library, London NW1 2DB, UK. [2]School of Mathematical Sciences, Queen Mary University of London, London E1 4NS, UK. [3]School of Mathematical Sciences, University of Southampton, Southampton SO17 1BJ, UK. [4]Faculty of Medicine, University of Southampton, Southampton SO17 1BJ, UK. [5]UCL Cancer Institute, University College London, London WC1E 6DD, UK. ✉e-mail: abaptista@turing.ac.uk; cbanerji@turing.ac.uk

understanding of intra-cellular interactions and instead rely on sparse graphical representations, known as biological networks, which can be weighted by biological samples to describe relevant dynamics[9]. The notion that a (weighted) network has an underlying geometry is well-studied and there are numerous methodologies for network embedding[14], with application to biological networks[15,16]. Recently, discrete analogues of tools from differential geometry[17,18], a rich mathematical field for studying manifolds and their curvatures, have been applied to the study of biological networks[19–23]. These tools provide a new window into the geometry of cell fate and a rich theoretical literature to apply.

In particular, discrete analogues of Ricci curvature, well known for its use to describe the curvature of space-time in Einstein's theory of general relativity, have been employed to discriminate biological networks weighted with cancer gene expression data from corresponding healthy tissue[19]. In 2015, Sandhu et al.,[19] proposed a theoretical link between network entropy and a discrete version of Ricci curvature (Ollivier-Ricci curvature[17]) computed over the edges of a weighted network. This link was motivated by the theoretical results of Lott and Villani, relating a lower bound of the Ricci curvature on a metric-measure space to the convexity of an entropy functional[24], suggesting that Ricci curvature and entropy (computed in this way) may be positively correlated. Though network entropy is not theoretically equivalent to the entropy functional from the metric-measure space setting, it was found that, like network entropy, total Ollivier-Ricci curvature is elevated on networks weighted with cancer data, compared to healthy[19]. Subsequently, similar results have been obtained, using the less computationally intensive Forman-Ricci curvature[21,23], including that this curvature decreases during cellular differentiation, again like network entropy. It is of note, however, that depending on the construction of this Forman-Ricci curvature, investigators have demonstrated both positive[22] and negative[25] correlations with network entropy.

Cellular differentiation and oncogenesis like all biological events are dynamic processes, and the recent results detailed above suggest that the geometry of the underlying space of intra-cellular interactions, described by biological networks, may change predictably during their progression. The dynamic evolution of manifolds is a well-studied topic in differential geometry. In a seminal contribution to the field, Hamilton introduced Ricci flow as a tool to study the topological implications of deforming a metric on a manifold according to its Ricci curvature[26], which led subsequently to the striking solution of the Poincaré conjecture by Perelman[27,28]. Like curvature Ricci flow can also be defined in a discrete setting[29], and recently discrete Ricci flows and curvatures have been applied to problems in network theory[30–32] such as network alignment[33], community detection[34–36], functional community inference for biological networks[37] and phase transitions in time-varying complex networks[38].

In what follows we first present some background on the computation of network entropy and discrete Ricci curvatures in the context of gene expression weighted protein-protein interaction networks. We then propose a framework for employing a discrete Ricci curvature and normalised Ricci flow to predict dynamic trajectories between temporally linked gene expression samples. We next consider the relationship between our Forman-Ricci curvature construction and network entropy; using a simple toy network we show that the two network measures are not always positively correlated. We find that in promiscuous signalling regimes (such as in stem cells) the measures do positively correlate, but in lower entropy regimes they may anti-correlate, suggesting the two measures are complementary rather than interchangeable. By analysing over 6000 single-cell transcriptomes, we confirm these propositions, demonstrating that network entropy and our Forman-Ricci curvature positively correlate in stem cells, but negatively correlate in cancerous and differentiated samples. Lastly, we consider two independent transcriptomic time courses describing

multiple time points during cellular differentiation in different tissues. Using our Ricci flow construction we derive gene expression trajectories from the first time point sample to the last, faithfully predicting the ordering of intermediate samples, without prior knowledge.

## Results

### Intuition, definitions and preliminaries

Intuitively, we interpret the Waddington Landscape as analogous to the phase space of transcriptomic dynamics during cellular differentiation (Fig. 1A). Let $n$ denote the number of genes in the genome and $\mathbf{x^t} := (x_i^t)_{i=1}^n > 0$ denote the vector of transcript abundance for each gene at time $t \in \mathbb{R}^+$. Consideration of $\frac{d\mathbf{x^t}}{dt}$ yields an $n$ dimensional phase space $\phi$, describing permissive trajectories of gene expression. Trajectories between two points in $\phi$ represent geodesics from one transcriptomic state to another, and distances along these trajectories can be computed by equipping $\phi$ with a Riemannian metric $g$. The degree to which these geodesic distances differ from Euclidean distances can be assessed via consideration of Ricci curvature, allowing us to recast the $n$ dimensional manifold $(\phi, g)$ as an $n+1$ dimensional manifold $\Phi$ with a Euclidean geometry. This added dimension allows us to interpret the "height" of Waddington's Landscape, and permits investigation of its association with cellular differentiation states.

A key issue in progressing this construct is the knowledge of $\frac{d\mathbf{x^t}}{dt}$, which will be a highly sophisticated function incorporating transcription, translation and degradation of mRNA and protein for each gene, as well as the complexities of epigenetic regulation, gene-regulatory networks, protein-protein interaction networks and cell-cell/micro-environment interactions.

It has been shown however, that integration of transcriptomic data with a protein-protein interaction network (PIN), compiled from multiple sources, yields an entropy rate which is a clear correlate of cellular differentiation potential and thus represents a proxy for "height" in Waddington's differentiation landscape[9]. This suggests a pragmatic approach considering $\frac{d\mathbf{x^t}}{dt}$ purely constructed from protein-protein interactions, may be sufficient for initial interrogation of the structure of $\Phi$, in lieu of a more rigorous theoretical understanding of other contributors.

### Network entropy and discrete Ricci curvature

In our construct we let $G = (V, E)$ denote the undirected graph describing the human PIN, with adjacency matrix $A = (a_{ij})_{i,j \in V}$, where $|V| = n$. For any $\mathbf{x} \in \phi$, where $x_i > 0$ for all $i \in V$, we define the weighted adjacency matrix $W(\mathbf{x}) = (a_{ij} x_i x_j)_{i,j \in V}$, and the row-stochastic matrix, $P(\mathbf{x}) = (p_{ij}(\mathbf{x}))_{i,j \in V}$, where:

$$p_{ij}(\mathbf{x}) = \frac{a_{ij} x_j}{\sum_{k \in V} a_{ik} x_k}. \tag{1}$$

The entropy rate $S_R(\mathbf{x})$ of $P(\mathbf{x})$ (hereafter denoted as network entropy, Methods) decreases as cells differentiate, this has been established in bulk and single-cell transcriptomic data from cells at different stages of differentiation and throughout differentiation time courses, by us and multiple independent investigators[9,11,22]. Network entropy is also higher in cancerous compared to healthy tissue, and is prognostic in breast and lung cancer[10,11,39].

Sandhu et al.,[19] proposed a positive correlation between network entropy and a discrete version of Ricci curvature computed over edges in a weighted network $(Ric_e(\mathbf{x}))_{e \in E}$, with network average or total Ricci curvature defined by:

$$Ric(\mathbf{x}) = \sum_{i \in V} \pi_i(\mathbf{x}) \frac{1}{\deg(i)} \sum_{j \in V} a_{ij} Ric_{(i,j)}(\mathbf{x}), \tag{2}$$

where $\deg(i) = \sum_{j \in V} a_{ij}$ and where $(\pi_i(\mathbf{x}))_{i=1}^n$ is the stationary distribution of $P(\mathbf{x})$. The correlation between network entropy and total discrete

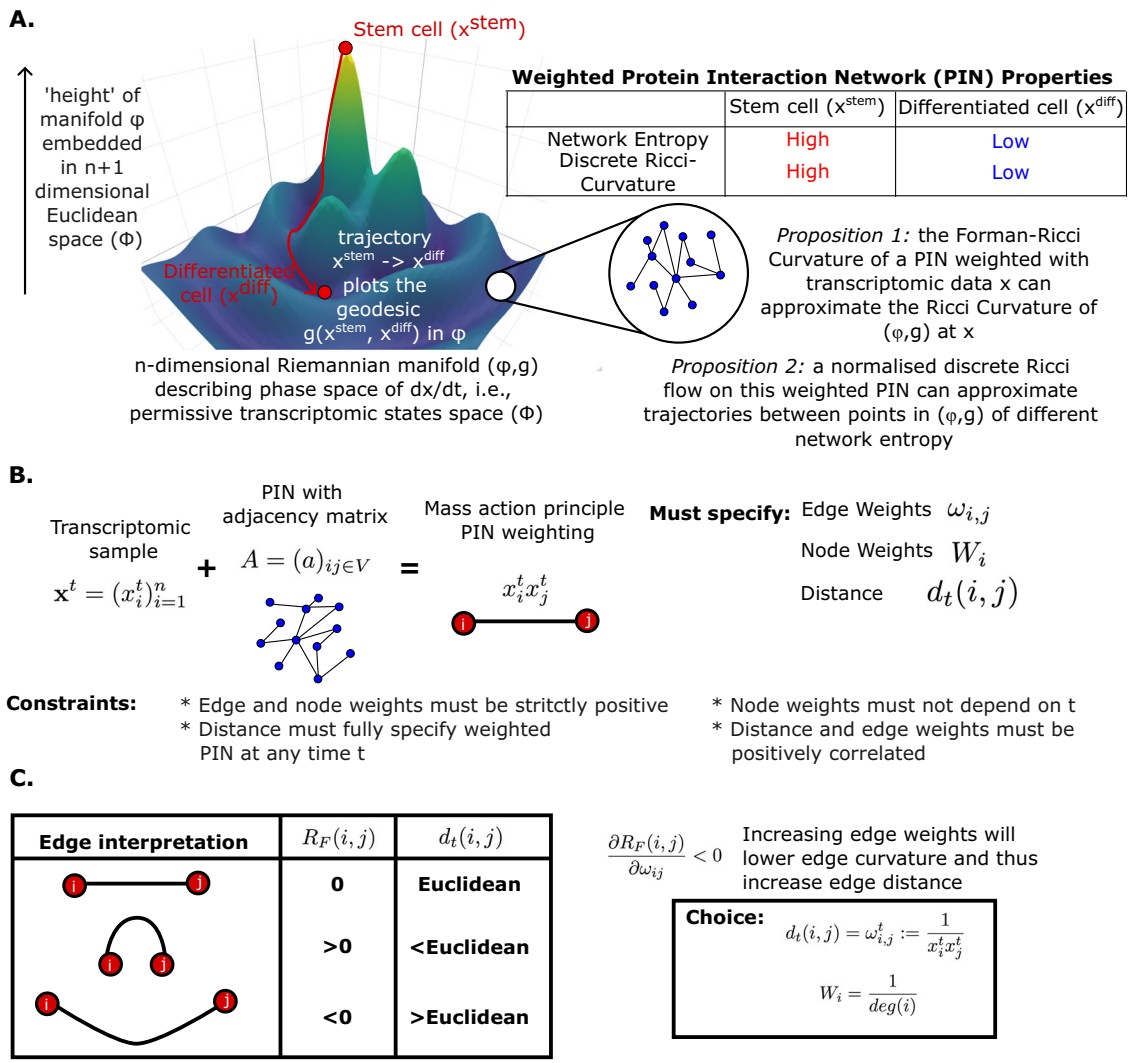

**Fig. 1 | Overview of Ricci curvature and flow approach for biological networks.** **A** Schematic of the transcriptomic phase space interpretation of the Waddington Landscape. Permissive trajectories are interpreted as geodesics which can be considered within Euclidean geometry via consideration of Ricci curvature. Protein interaction networks can be used to approximate height in the landscape by network entropy and possibly by discrete Ricci curvature. **B** Schematic describing mass action principle weighting of protein interaction network with transcriptomic data, alongside required parameter choices and corresponding constraints to implement Ricci flow. **C** An interpretation of edge curvature in terms of node proximity in an underlying metric space and consequences for parameter choices.

Ricci curvature has since been considered by several studies in the following form[19,21,22]:

$$\Delta S_R(\mathbf{x_t})\Delta Ric(\mathbf{x_t}) \geq 0. \tag{3}$$

While not an unreasonable deduction, justification for this inequality derives from a theoretical investigation of metric-measure spaces $(M, d, m)$, where $(M, d)$ is a metric space and $m$ is a measure on the Borel $\sigma-$algebra of $M$ (Methods)[24,40]. Investigation in this setting uncovered a relationship between the convexity of a relative entropy, computed over the space of probability measures on $(M, d)$, with respect to the measure $m$ and a lower bound of the Ricci curvature of $(M, d, m)$[24,40]. From this association, it was concluded that the negative of the relative entropy and Ricci curvature are positively correlated[19]. We note, however, that the network setting is not equivalent to metric-measure spaces. In particular network entropy (an entropy rate) is not equivalent to the relative entropy described by[24]. The inequality (3) is therefore not guaranteed from the results on metric-measure spaces[24,40].

Moreover, discrete Ricci curvatures, though often theoretically rich, are not exact quantifiers of the continuous Ricci curvature on a manifold. There are several approaches to computing a discrete Ricci curvature on edges of a network, including Ollivier-Ricci curvature[17] and Forman-Ricci curvature[18], both of which have been applied to biological networks and demonstrate elevated total curvature in cancer[19,21,22]. Forman-Ricci curvature follows a combinatorial construction as follows:

$$R_F(i,j) = W_i + W_j - (\omega_{ij})^{1/2}\left[W_i\sum_{k \neq j}(\omega_{ik})^{-1/2} + W_j\sum_{k \neq i}(\omega_{kj})^{-1/2}\right] \tag{4}$$

where $(W_i)_{i \in V}$ is a vector of vertex weights and $(\omega_{ij})_{(i,j) \in E}$ is a vector of edge weights. We note that Forman-Ricci curvature is less computationally intensive to evaluate than Ollivier-Ricci curvature.

Though the discrete entropy and curvature measures do not exactly correspond to the metric-measure space setting, the relation (3) suggests an intriguing geometrical interpretation for the observation that network entropy decreases during cellular differentiation. Transcriptomic states representing undifferentiated cells $\mathbf{x^{stem}} \in \phi$, have higher network entropy compared to differentiated cells $\mathbf{x^{diff}} \in \phi$. Under (3) it follows that $Ric(\mathbf{x^{stem}}) > Ric(\mathbf{x^{diff}})$. Tree-like networks have a

very low curvature, whereas cliques are highly curved[33], giving a natural interpretation to this inequality in terms of more deterministic pathway activation during differentiation.

In our phase space analogy to Waddington's Landscape, with $\frac{d\mathbf{x^t}}{dt}$ essentially described by $W(\mathbf{x^t})$, we see stem cells occupying regions of high curvature (hill tops) and curvature decreasing as cells differentiate, analogously, rolling downhill to valleys. This gives us an intuitive, empirical tool to understand construction of the $n+1$ dimensional space $\Phi$ for the $n$ dimensional phase space $(\phi, g)$ at given data points.

## Normalised discrete Ricci flow

Cellular differentiation is a dynamic process and typically we only have data for start and end points $\mathbf{x^{stem}}$ and $\mathbf{x^{diff}}$ and perhaps a handful of points between. We consider extrapolation between these data points via a discrete normalised Ricci flow.

We propose to use a discrete version of the 2-dimensional normalised Ricci flow, which has previously been considered in the context of weighted networks[30]:

$$d_{t+\Delta t}(i,j) = d_t(i,j) + \Delta t(Ric(\mathbf{x^t})_{(i,j)} - \overline{Ric}_{(i,j)})d_t(i,j) \quad (5)$$

for $\Delta t > 0$, where $d_t(i,j)$ is a distance between connected nodes $i,j \in V$ at time $t$, $Ric(\mathbf{x^t})_{(i,j)}$ is the Ricci curvature on edge $(i,j) \in E$ at time $t$ and $\overline{Ric}_{(i,j)}$ is an edge-wise normaliser to which we want to converge.

Here we consider $t = 0$ to refer to the undifferentiated cell state $\mathbf{x^{stem}}$ and define the normaliser via the fully differentiated state: $\overline{Ric}_{(i,j)} = Ric(\mathbf{x^{diff}})_{(i,j)}$. We postulate that (5) will permit estimation of a permissive trajectory from $\mathbf{x^{stem}}$ to $\mathbf{x^{diff}}$ in $\phi$.

For (5) to generate trajectories the following properties are required (Fig. 1B):

- Knowledge of $d_t(i,j)$ must be sufficient to calculate $Ric(\mathbf{x^t})_{(i,j)}$.
- $\Delta t$ must be sufficiently small to prevent negative values of $d_t$. The following properties are also desired:
- Knowledge of $d_t$ allows calculation of $\mathbf{x^t}$ or some transformation thereof, e.g., $W(\mathbf{x^t})$. This will permit comparison to intermediate real data points to validate the approach.
- Computation time of Ricci curvatures must be sufficiently short to permit multiple iterations rapidly, as for large PINs such as those investigated here, there are typically ~150,000 edges.

In what follows we compute $Ric(\mathbf{x^t})_{(i,j)}$ as a Forman-Ricci curvature $R_F^t(i,j)$ with edge weights $\omega_{ij} := \omega_{ij}^t = \frac{a_{ij}}{x_i^t x_j^t}$ and node weights $W_i = \frac{1}{\deg(i)}$. $Ric(\mathbf{x^t})_{(i,j)} = R_F^t(i,j)$ thus obeys:

$$R_F^t(i,j) = \deg(i)^{-1} + \deg(j)^{-1} - (x_i^t x_j^t)^{-1/2}$$
$$\left[ \deg(i)^{-1} \sum_{k \neq j} (a_{ik} x_i^t x_k^t)^{1/2} + \deg(j)^{-1} \sum_{k \neq i} a_{ik}(a_{kj} x_k^t x_j^t)^{1/2} \right] \quad (6)$$

We further choose $d_t(i,j) = \omega_{ij}^t$. These choices satisfy all of our required and desired properties and detailed justification can be found in the Methods.

## Positive correlation between network entropy and total Forman-Ricci curvature requires a specific signalling regime

Previous studies have demonstrated a positive correlation between network entropy and network average (or total) discrete Ricci curvature computed on differentiating stem cells[19,22]. However, recently it has been demonstrated using a slightly different construction of Forman-Ricci curvature that a negative correlation can be observed with network entropy[25]. As discussed above a positive correlation between network entropy and discrete Ricci curvature is not guaranteed in general, as the motivating theoretical results relate to slightly different quantities[24,40].

To gain intuition we investigated the association between our version of Forman-Ricci curvature and network entropy on a simple $k$-star network displayed in Fig. 2A, consisting of $k+1$ nodes, of which $k$ have a single edge connecting them to a central node $i$. We assign each node $l \neq j$ a weight $x_l = 1$ and assign node $j$ a weight $x_j = \epsilon > 0$. We can derive analytical expressions for network entropy ($S_R$) and total Forman-Ricci curvature ($R_F$, defined by (6) and (2)) on this simple network in terms of $k$ and $\epsilon$ (Methods).

We performed a numerical analysis of these expressions for various values of $k \in \mathbb{Z}^+ \setminus 1$ and $\epsilon > 0$ (Fig. 2B–D). By construction $S_R$ is maximal for $\epsilon = 1$, regardless of $k$. For $k = 2$, $R_F$ also has a global maximum at $\epsilon = 1$ and the positive correlation with $S_R$ expressed in (3) holds. However for all other values of $k$, the association between network entropy and total Forman-Ricci curvature follows two regimes depending on $\epsilon$ (Fig. 2D). For $\epsilon < 1$ (3) holds and network entropy and total Forman-Ricci curvature are positively correlated. However, for $\epsilon > 1$ we can always find a range of values of $\epsilon$ for which network entropy and total Forman-Ricci curvature are negatively correlated, this range becomes larger as $k$ increases.

Though these results only apply to a very simple network, they suggest a fundamental difference in what network entropy and total Forman-Ricci curvature are measuring. This suggests these measures are complementary, rather than interchangeable as has been previously proposed[19]. In our simple network, network entropy is maximised for $\epsilon = 1$. We can reduce network entropy by reducing $\epsilon$, signalling more the $k-1$ neighbours of our central node $i$ at the cost of reducing signalling to our chosen neighbour $j$, a strategy we call "many for one" (Fig. 2E), in this case $R_F$ will also decrease. Alternatively, we can reduce network entropy by increasing $\epsilon$, and signal more to our chosen node $j$ at the cost of signalling less to our remaining neighbours, a strategy we call "one for many" (Fig. 2E), in this case for larger values of $k$, $R_F$ may increase.

Network entropy is blind to the two signalling strategies, but they are biologically distinct. The "one for many" strategy mirrors deterministic pathway activation, characteristic of a low entropy regime. This strategy is more likely in a highly committed cell, performing a very specific function[9]. Variation in gene expression amongst well-differentiated cells may therefore capture the negative correlation between network entropy and total curvature we have demonstrated possible by our theoretical investigation. Conversely, the "many for one" signalling strategy, though not maximising entropy, represents a more disordered state than the "one for many" strategy, maintaining the possibility of diverse pathway activation without committing. This regime mirrors the promiscuous signalling of stem cells, which must maintain the option to differentiate and perform a wide variety of functions[9]. Variation in gene expression amongst stem cells may therefore capture the positive correlation between network entropy and total curvature, which we have theoretically demonstrated more dominant in "many for one" signalling.

## The degree of correlation between network entropy and total Forman-Ricci curvature has biological relevance

Our theoretical results suggest that our $S_R$ and $R_F$ may be positively correlated in stem cells, but negatively correlated in more differentiated tissue. Previous studies reporting an association between network entropy and total Forman-Ricci curvature typically present results on stem cell populations[21,22,25]. Though the curvatures of more differentiated and cancerous tissues are often also examined, the association with network entropy in these tissues is typically not reported[25,41]. We note that these studies also employ slightly different constructions of Forman-Ricci curvature than our own and while most show a positive correlation with network entropy in stem cells[19,22], one shows a negative correlation[25].

We analysed the previously considered scRNAseq data sets of Chu et al.[11,22,25,42] describing the early stages of embryonic stem cell (ESC)

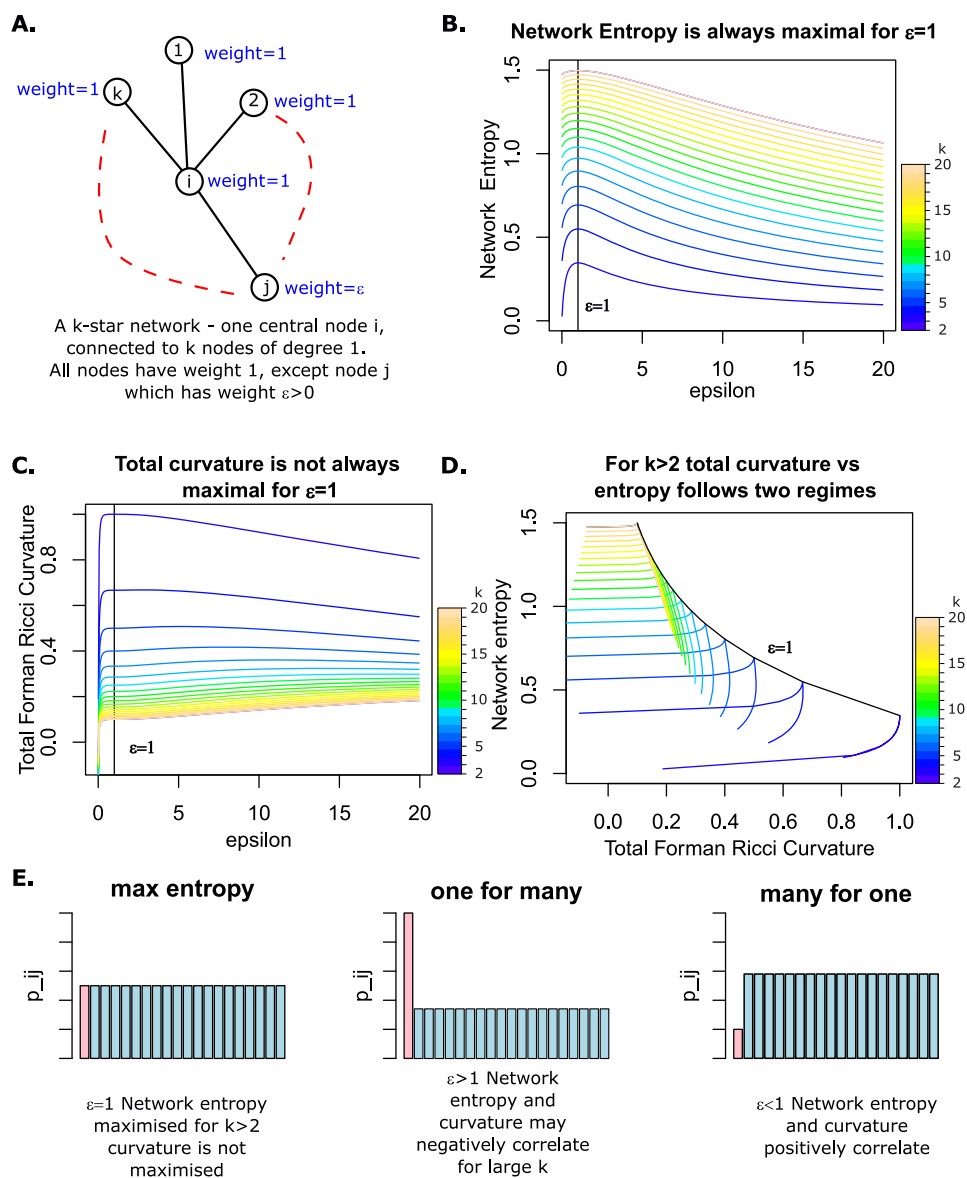

**Fig. 2 | Examining the association between network entropy and total Forman-Ricci curvature in a toy network. A** A simple *k*-star network with *k*+1 nodes, *k* nodes of degree 1 are connected to a central node of degree *k*. Node weights are set to 1 for all nodes except *j*, which is set to some $\epsilon > 0$. **B** Plot of network entropy against $\epsilon$ for *k* = 2, ..., 20, colours from blue to brown denote increasing *k*. **C** Plot of

total Forman-Ricci curvature against $\epsilon$ for *k* = 2, ..., 20, colours from blue to brown denote increasing *k*. **D** Plot of total Forman-Ricci curvature against network entropy for *k* = 2, ..., 20, colours from blue to brown denote increasing *k*. **E** Probability bar charts comparing the "one for many" to "many for one" regimes.

differentiation. These data consist of 2 separate experiments, one describing 1018 single cells assayed at different stages of multipotency and a second describing 758 single cells assayed at 6 distinct time points during ESC differentiation. On both these data sets we found that network entropy and our total Forman-Ricci curvature were positively correlated (Pearson's $r > 0.78$, $p < 2.2 \times 10^{-16}$) and discriminate distinct lineages during stem cell differentiation (Fig. 3A, B) as previously reported[11,22,25].

We next analysed a large scRNAseq data set describing 1257 malignant and 3256 healthy single cells from 19 patients with malignant melanoma[43], on which total curvature values have previously been calculated, but the association with network entropy was not presented[22,25]. These cells represent more differentiated tissue and as hypothesised from our theoretical investigation, we found a negative association between network entropy and our total Forman-Ricci curvature on these cells (Pearson's $r = -0.77$, $p < 2.2 \times 10^{-16}$, Fig. 3C). We also found that malignant cells displayed higher values of network

entropy as expected (two-tailed Wilcoxon $p < 2.2 \times 10^{-16}$)[9], however, they displayed lower values of total Forman-Ricci curvature (two-tailed Wilcoxon $p < 2.2 \times 10^{-16}$, Fig. 3C). Considering healthy and malignant cells separately, we found that the correlation between network entropy and total Forman-Ricci curvature was significantly more negative across healthy cells compared to malignant (control cells: Pearson's $r = -0.83$, $p < 2.2 \times 10^{-16}$), malignant cells: Pearson's $r = -0.009$, $p = 0.76$, Fisher's $z$-transformation: $p < 2.2 \times 10^{-16}$).

To confirm this finding we analysed an independent data set describing 272 malignant and 160 healthy cells from patients with colorectal cancer[25,44]. We again identified a negative correlation between network entropy and total Forman-Ricci curvature (Pearson's $r = -0.86$, $p < 2.2 \times 10^{-16}$, Fig. 3D), with higher network entropy (two-tailed Wilcoxon $p = 1.5 \times 10^{-6}$) but lower total Forman-Ricci curvature (two-tailed Wilcoxon $p = 8.0 \times 10^{-4}$) in cancerous cells. Again, considering healthy and malignant cells separately, the correlation between network entropy and total Forman-Ricci curvature was

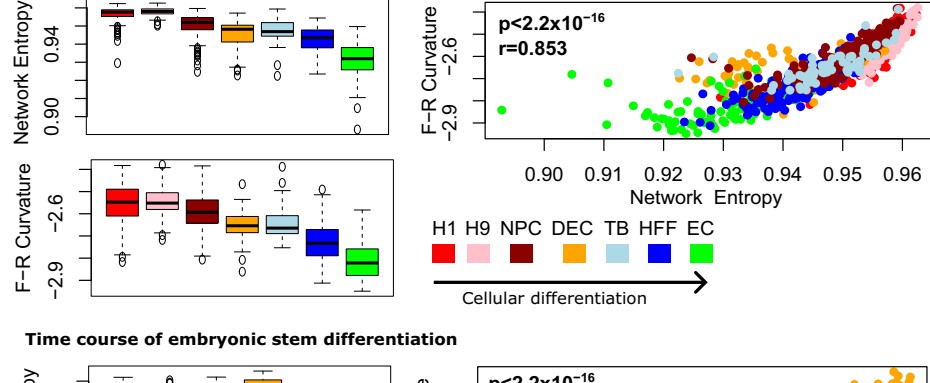

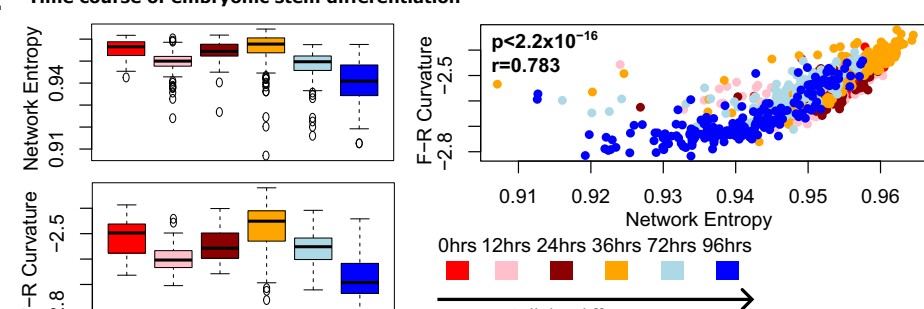

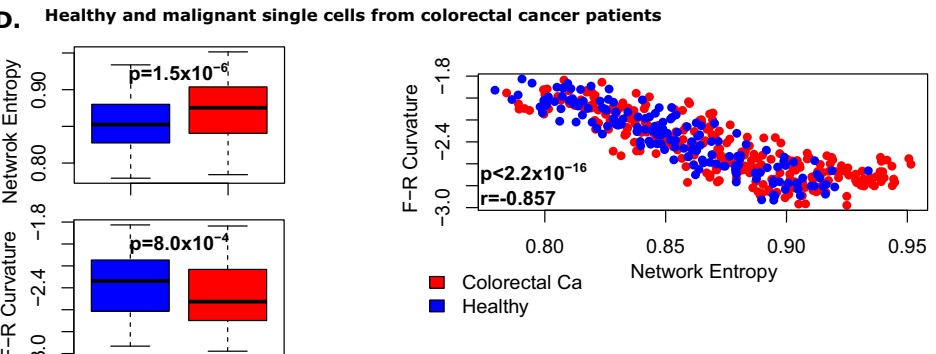

**Fig. 3 | Forman-Ricci curvature and network entropy follow two distinct regimes in biological data.** Boxplots and scatter plots display network entropy and Forman-Ricci curvature for **A.** 1018 single cells during distinct stages of embryonic stem cell (ESC) differentiation (colours label samples at different stages of differentiation: red=H1, pink=H9, dark red=NPC, orange=DEC, light blue=TB, dark blue=HFF, green=EC), **B** 758 single cells sampled at 6 distinct time points of ESC differentiation (colours label samples taken at different times during differentiation: red=0 h, pink = 12 h, dark red = 24 h, orange = 36 h, light blue = 72 h, dark blue = 96 h), **C** 1257 malignant and 3256 control cells from 19 patients with malignant melanoma (colours label phenotype: red=malignant, blue=healthy) and **D** 272 malignant and 160 healthy cells from patients with colorectal carcinoma (colours label phenotype: red=colorectal cancer, blue=healthy). Boxplots present data as follows: minima: minimum value, maxima: maximum value, centre: median, bounds of box: first and third quartile, whiskers: lowest value within 1.5 × interquartile range of the first quartile, to largest value within 1.5 × interquartile range of the third quartile. Two-sided Wilcoxon *p*-values are displayed on boxplots, and Pearson's *r* and corresponding two-sided *p*-values are displayed on scatter plots. In stem cells network entropy and total Forman-Ricci curvature positively correlate, while for more committed cells there is a negative correlation. Source data are provided as a Source Data file.

significantly more negative across healthy cells compared to malignant, though the difference was more subtle than in the melanoma data set (control cells: Pearson's $r = -0.90$, $p < 2.2 \times 10^{-16}$, malignant cells: Pearson's $r = -0.83$, $p < 2.2 \times 10^{-16}$, Fisher's $z$-transformation $p < 4.8 \times 10^{-3}$).

This suggests that network entropy and total Forman-Ricci curvature are not interchangeable measures of cell potency, but complementary. Increasing network entropy is seen in both less differentiated tissue and cancer, while total Forman-Ricci curvature increases in less differentiated tissue and decreases in cancer. Together these measures present a more complete picture of the global intra-cellular signalling state.

### Ricci flow for approximating transcriptomic trajectories

We have found that network entropy and our total Forman-Ricci curvature are related quantities but not interchangeable.

We next consider whether Ricci flow can approximate realistic trajectories through gene expression phase space during cellular differentiation. We first considered the time course scRNAseq data set of Chu et al.[42], describing ESC differentiation at 6 time points. For each time point we computed the mean transcriptomic vector across single cells, which we considered representative of the transcriptomic state at this time point, giving us a set of 6 vectors $(\mathbf{x^t})_{t=0}^{5}$ (Fig. 4A). To provide a null model we considered a straight line trajectory from $W(\mathbf{x_0})$ to $W(\mathbf{x_5})$ (Methods). We computed the Euclidean distance between points along this straight line and the true intermediate data points $(W(\mathbf{x_t}))_{t=1}^{4}$, to determine the ordering of the true data points along the straight line trajectory (Methods, Fig. 4B). As anticipated the straight line trajectory did not pass the true data points in the correct order, and the distance along the trajectory to the closest pass of the true data point was not correlated with differentiation time of the true data point (Pearson's $r = 0.85$, $p = 0.153$, Fig. 4C). We next considered the trajectory from $W(\mathbf{x_0})$ to $W(\mathbf{x_5})$ produced by our normalised discrete Ricci flow described by (5) (Methods). We found that the Ricci flow trajectory passed by the true data points in the correct order, and the number of iterations to the closest pass of the true data points correlated with the differentiation time of those points (Pearson's $r = 0.96$, $p = 0.04$, Fig. 4D).

To confirm the finding that Ricci flows correctly orders differentiation trajectories, we considered our data set of bulk RNA-sequencing of human myoblast differentiation into multinucleated myotubes, with transcriptomic samples taken at 8-time points in triplicate (Fig. 5A)[45]. Performing analysis as above, separately for each triplicate, we found that closest pass progression along a null model linear trajectory correlated with differentiation time but could not robustly discriminate time points across triplicates (Pearson's $r = 0.79$, $p = 1.0 \times 10^{-4}$, Fig. 5B). In contrast, closest pass Ricci flow iterations were highly correlated with differentiation time (Pearson's $r = 0.93$, $p = 3.7 \times 10^{-8}$, Fig. 5B) and were tightly reproducible across triplicates, discriminating all time points, with the exception of the first two intermediate time points. These initial time points were taken only 90 min apart and thus are unlikely to represent a significant dynamic change.

## Discussion

Numerous measures have been developed in network theory to analyse network properties. Classic approaches include studying the degree distribution, clustering coefficient, and shortest path between nodes, all of which provide insights into the network's geometry[46]. However, to study the geometric and topological properties of networks more deeply, discrete adaptations of differential geometry have become widely applied[19,22,30–33,36,41]. In differential geometry curvature is a key actor, describing the local behaviour of a manifold, and geometric flows can be employed to perturb this important property and examine the consequences. By treating networks as discrete counterparts of manifolds, we can view them as geometric objects and discrete curvatures and flows on networks have proven effective tools for addressing common network theory questions[31–33,36].

Here we investigated discrete Ricci curvature and Ricci flow, to study properties of biological signalling in differentiating and malignant cells. This work builds on the finding that network entropy is a proxy for "height" in Waddington's Landscape— having higher values on stem cells and malignant cells compared to healthy differentiated tissue[9–11]—by investigating the enticing theoretical link between Ricci curvature and entropy[19,24,40]. We propose a framework to calculate the total Forman-Ricci curvature of a single biological sample, which is compatible with a discrete Ricci flow, to infer trajectories between the intra-cellular signalling regimes of two temporally connected transcriptomic samples.

By investigating our framework in a simple analytically tractable setting, we prove that network entropy and our total Forman-Ricci curvature are not guaranteed to be positively correlated. Our investigation suggests that a positive correlation is likely across samples with a highly promiscuous signalling regime (such as stem cells), with a negative correlation more likely across cells with deterministic signalling (differentiated cells). We provide empirical evidence for this theoretical hypothesis through the analysis of > 6000 single-cell transcriptomes. Interestingly, we found that cancer cells have a higher network entropy but lower total Forman-Ricci curvature than healthy differentiated cells and that the correlation between network entropy and total Forman-Ricci curvature is less negative in cancerous cells compared to healthy. This is in contrast to stem cells where both network entropy and total Forman-Ricci curvature are higher than healthy differentiated cells and positively correlated.

One of the hallmarks of malignancy is anaplasia—the de-differentiation of cancerous cells compared to their healthy counterparts. Anaplasia is typically quantified by histological grade, where tumour cells are compared morphologically to their healthy counterparts and assigned a low grade if they appear similar, or a high grade if they have lost the appearance associated with specialised function. Anaplastic malignant cells gain some of the hallmarks of stem cells, such as a higher proliferative capacity, they also gain additional functions, including those which facilitate metastasis. Our theoretical results suggest that the loss of negative correlation between network entropy and total Forman-Ricci curvature in malignant cells may represent an increase in "many to one" signalling compared to healthy cells, expected in anaplasia. Highly anaplastic cells may even attain a signalling regime more characteristic of stem cells, and show a positive correlation between network entropy and total Forman-Ricci curvature.

By applying our normalised discrete Ricci flow to the first and last time point of time courses of cellular differentiation from two distinct tissues, we derived biological network rewiring trajectories, which accurately predicted intermediate time points. Predictions made by this approach require experimental validation but offer the possibility of deeper insights into the molecular events underpinning cellular differentiation and early biomarker detection for malignancy and regenerative pathology.

Our findings contrast with other studies, which proposed a positive correlation between network entropy and total discrete curvature of a biological network, by appealing to results on metric-measure spaces[19,24,40]. There are a number of reasons for this contrast. Firstly, the discrete network setting is not the exact analogue of the metric-measure space setting and in particular the definitions of "entropy" in the two settings are not identical. Secondly, discrete approximations of Ricci curvature for networks are non-unique and there are several ways of defining them depending upon context, including Ollivier-Ricci curvature derived from optimal transport considerations[17] and Forman-Ricci curvature derived from consideration of cell complexes[18]. It has been shown that node averages of these different

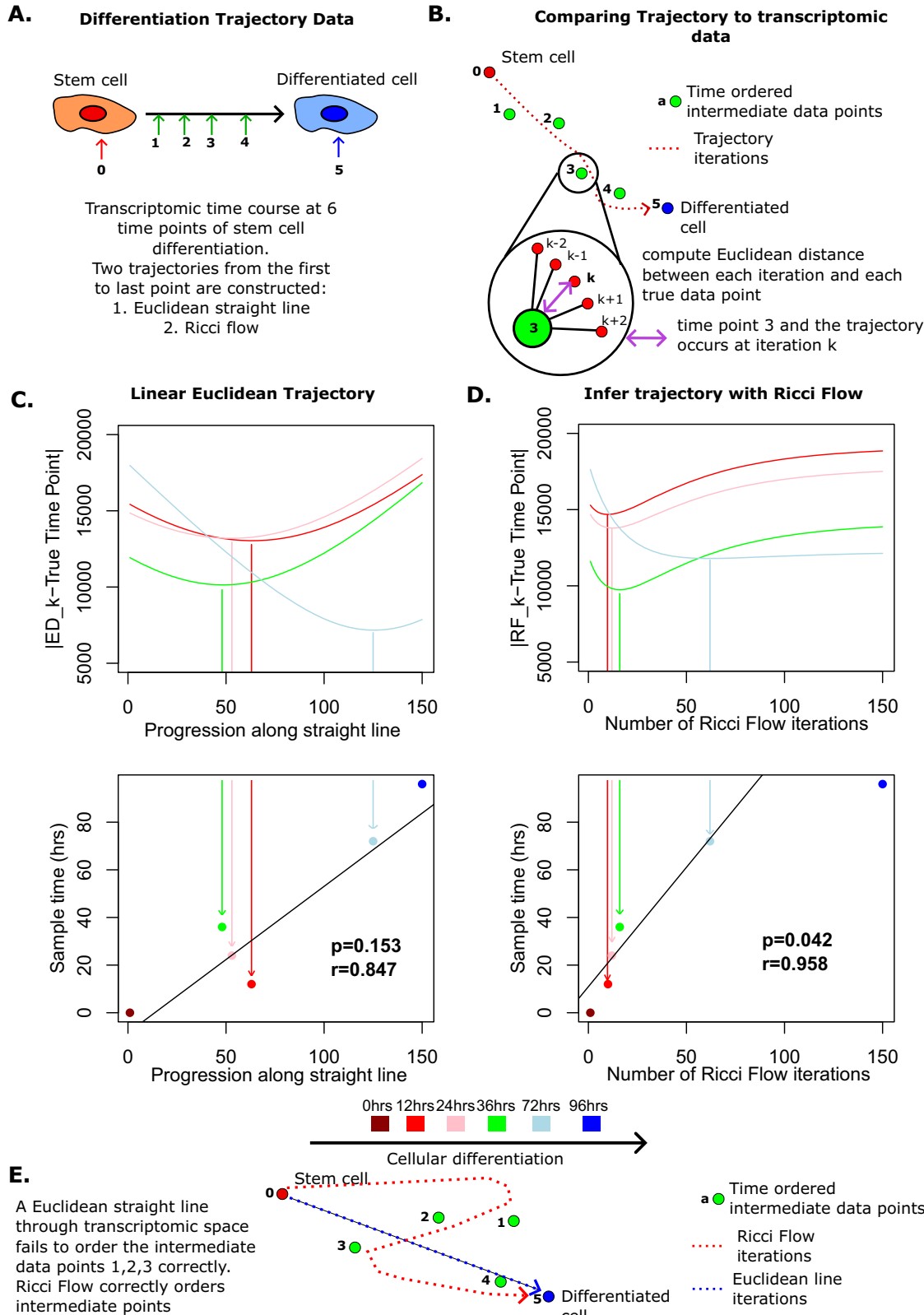

**Fig. 4 | Ricci flow correctly orders differentiation time course during embryonic stem cell (ESC) differentiation. A** Schematic shows samples taken during ESC differentiation. **B** Schematic demonstrates how transcriptomic trajectories are compared to true biological data points. **C** Ordering of data points along a straight Euclidean trajectory, two-sided *p*-value and Pearson's *r* describe the association between closest pass iteration of the trajectory to true data points and differentiation time of the true data points. **D** Ordering of data points along Ricci flow trajectory, two-sided *p*-value and Pearson's *r* describe the association between

closest pass iteration of the trajectory to true data points and differentiation time of the true data points. The final time point, by construction occurs at iteration 150 at the end of the trajectory. Points and lines are coloured according to differentiation time of the sample being assessed for closest pass: dark red = 0 h, red = 12 h, pink = 24 h, green = 36 h, light blue = 72 h, dark blue = 96 h. **E** Schematic compares Euclidean straight line to the Ricci flow trajectory. Source data are provided as a Source Data file.

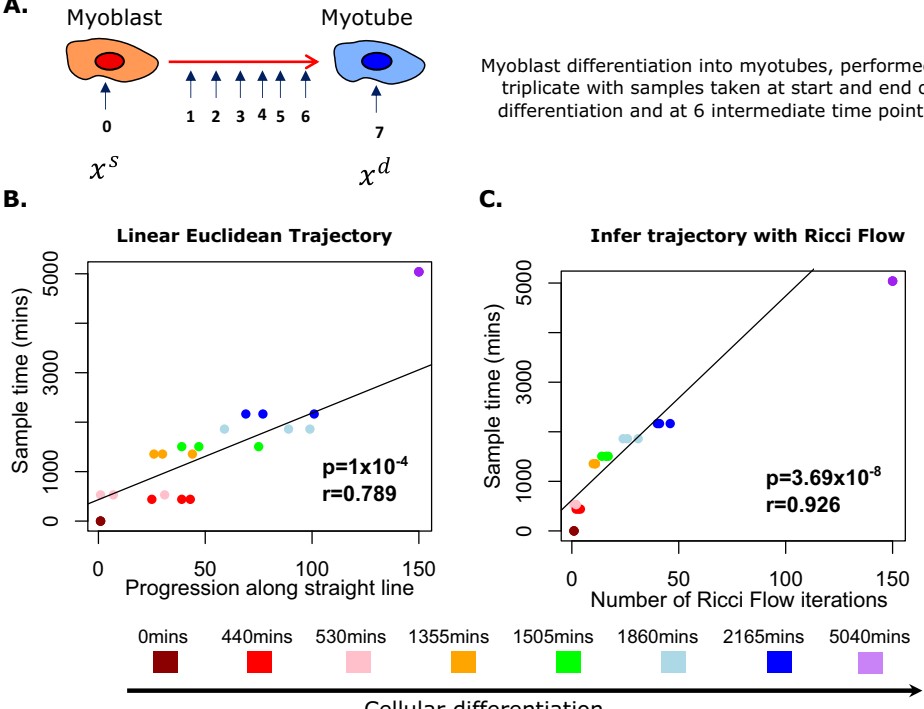

**Fig. 5 | Ricci flow correctly orders differentiation time course during myoblast differentiation. A** Schematic shows samples taken during myoblast differentiation in triplicate. **B** Ordering of data points along Euclidean straight line, two-sided *p*-value and Pearson's *r* describe the association between closest pass iteration of the trajectory to true data points and differentiation time of the true data points. **C** Ordering of data points along Ricci flow trajectory, two-sided *p*-value and Pearson's *r* describe the association between the closest pass iteration of the trajectory to true data points and differentiation time of the true data points. The final time point, by construction, occurs at iteration 150 at the end of the trajectory. Points are coloured according to differentiation time of the sample being assessed for closest pass: dark red = 0 min, red = 440 min, pink = 530 min, orange = 1355 min, green = 1505 min, light blue = 1860 min, dark blue = 2165 min, purple = 5040 min. Source data are provided as a Source Data file.

discrete Ricci curvatures computed on the same network do not always correlate[20]. Moreover, if we focus only on the Forman-Ricci curvature employed here, it can be seen from (4) that there is considerable flexibility in its definition, via the selection of node and edge weights. Indeed a positive correlation between Forman-Ricci curvature and network entropy[22], became negative across the same samples when the investigators used a different choice of edge weights[25]. The selection of weights for Forman-Ricci curvature therefore requires careful consideration to ensure it is matched to context. In particular, it may be possible to choose weights which artificially engineer a correlation between total Forman-Ricci curvature and network entropy. Moreover, if we define both node and edge weights as variables which change temporally, as has been done previously[22,25], then a Ricci flow on edges as we have constructed is computationally intractable. Our findings therefore motivate theoretical investigation into how to translate the deep results from metric-measure spaces into the biological network setting with more fidelity, as well as a more robust understanding of the impact of parameter choices when applying Forman-Ricci curvature to weighted biological networks. Here we provide a framework for such theoretical investigation and show that our Forman-Ricci curvature is an informative biological network measure, complementing rather than simply correlating with network entropy by providing robust discrimination between healthy, cancerous and stem cells.

Our work paves the way towards addressing questions related to the prediction of network evolution over time and their study with tools adapted from differential geometry. Though both theoretical and experimental investigations are required to fully exploit this area, we demonstrate that important insights into the molecular mechanisms of health and disease can be achieved through analysis of discrete Ricci curvatures and flows.

## Methods
### Network entropy calculation
The computation of network entropy was as previously described[9–11] employing the SCENT package in R and the symmetric PIN compiled from multiple sources in 2016 available at https://github.com/aet21/SCENT. We denote the adjacency matrix of the PIN by $A = (a_{ij})_{i,j=1}^n$.

For each gene expression sample, genes were matched to proteins in the PIN, when multiple genes were mapped to a single protein, expression levels were averaged over and only the largest connected component of the PIN was considered post-matching. For each matched sample $\mathbf{x} = (x_i)_{i=1}^n > 0$ a weighted network $W(\mathbf{x}) = (a_{ij}x_ix_j)_{i,j\in V}$, and row-stochastic matrix, $P(\mathbf{x}) = (p_{ij}(\mathbf{x}))_{i,j\in V}$, where:

$$p_{ij}(\mathbf{x}) = \frac{a_{ij}x_j}{\sum_{k\in V}a_{ik}x_k} \tag{7}$$

were constructed.

We define the local entropy of node $i$ as

$$S_i(\mathbf{x}) = -\sum_{k\in V}p_{ik}(\mathbf{x})\log p_{ik}(\mathbf{x}) \tag{8}$$

the entropy rate associated with $P(\mathbf{x})$ is then given by

$$S_R(\mathbf{x}) = \sum_{i\in V}\pi_i(\mathbf{x})S_i(\mathbf{x}). \tag{9}$$

Where $\pi(\mathbf{x}) = (\pi_i(\mathbf{x}))_{i \in V}$ is the stationary distribution of $P(\mathbf{x})$ satisfying

$$\pi(\mathbf{x}) = P(\mathbf{x})\pi(\mathbf{x}). \tag{10}$$

As $G$ is undirected and a single connected component, by the Perron-Frobenius theorem the stationary distribution $\pi$ has an analytical solution given by:

$$\pi_i(\mathbf{x}) = \frac{\sum_{k \in V} a_{ij} x_i x_j}{\sum_{k,j \in V} a_{kj} x_k x_j}. \tag{11}$$

When presented in figures network entropy was calculated as the above entropy rate $S_R(\mathbf{x})$ normalised by the maximal entropy rate possible from the topology of the matched PIN, following our prior convention, to allow comparison across different networks[10,11].

## Construction of the Ricci flow equation

Formally for a smooth manifold $Y$ a Ricci flow defines for an open interval $(a,b) \in \mathbb{R}^+$ a Riemannian metric $d_t$ such that:

$$\frac{\partial d_t}{\partial t} = -2Ric(d_t) \tag{12}$$

the constant $-2$ is largely conventional and can be replaced with any $k < 0$, to ensure existence of a unique solution in finite time. Normalised Ricci flows are typically employed for convergence studies when certain properties, e.g., volume, are required to be finite

$$\frac{\partial d_t}{\partial t} = -2Ric(d_t) + \overline{Ric} \tag{13}$$

where $\overline{Ric}$ is a normaliser.

In 2 dimensions normalised Ricci flow is well-studied theoretically[47] and takes a special form:

$$\frac{\partial d_t}{\partial t} = (Ric(d_t) - \overline{Ric})d_t \tag{14}$$

For normalised discrete Ricci Flow we employ the following expression described in the main text and applied previously[21]:

$$d_{t+\Delta t}(i,j) = d_t(i,j) + \Delta t(Ric(\mathbf{x}^t)_{(i,j)} - \overline{Ric}_{(i,j)})d_t(i,j) \tag{15}$$

for $\Delta t > 0$, where $d_t(i,j)$ is a distance between connected nodes $i,j \in V$ at time $t$, $Ric(\mathbf{x}^t)_{(i,j)}$ is the Ricci curvature on edge $(i,j) \in E$ at time $t$ and $\overline{Ric}_{(i,j)}$ is an edge-wise normaliser to which we want to converge.

We next must choose expressions for $d_t(i,j)$ and $Ric(\mathbf{x}^t)_{(i,j)}$ which satisfy our required and desired properties outlined in the Results.

We select $Ric(\mathbf{x}^t)_{(i,j)}$ to be a Forman-Ricci curvature $R_F^t(i,j)$, as this discrete form of Ricci curvature is fast to compute compared to other versions such as Ollivier-Ricci curvature, and we must compute ~150,000 edge-wise curvatures per iteration of our Ricci flow. We choose the edge weights of this curvature to be $\omega_{ij} := \omega_{ij}^t = \frac{a_{ij}}{x_i^t x_j^t}$ and node weights $W_i = \frac{1}{\deg(i)}$. $R_F^t(i,j)$ thus obeys:

$$R_F^t(i,j) = \deg(i)^{-1} + \deg(j)^{-1} - (x_i^t x_j^t)^{-1/2}$$
$$\left[ \deg(i)^{-1} \sum_{k \neq j} (a_{ik} x_i^t x_k^t)^{1/2} + \deg(j)^{-1} \sum_{k \neq i} a_{ik} (a_{kj} x_k^t x_j^t)^{1/2} \right]. \tag{16}$$

We also choose $d_t(i,j) = \omega_{ij}^t$. We note that, as for other discrete Ricci flow studies[30,33], $d_t(i,j)$ is not a metric, as it fails the triangle inequality, however, it is small, implying "close proximity" of connected vertices $i,j \in V$ if the corresponding transcript levels of genes $i$

and $j$ are high at time $t$. In addition at each iteration of (5), this choice of $d_t(i,j)$ allows computation of $(\omega_{ij}^{t+\Delta t})_{(i,j) \in E}$, which can be input into (4), allowing computation of $(R_F^{t+\Delta t}(i,j))_{(i,j) \in E}$ and thus the next iteration of (5). This iterated $d_{t+\Delta t}$ can simply be inverted to give $W(\mathbf{x^{t+\Delta t}})$ which allows direct comparison of the Ricci flow generated transcriptomic distribution with real biological data. Our choice of $d_t$ thus satisfies all our desired properties and is a reasonable distance measure.

$W_i$ is chosen to be independent of $\mathbf{x^t}$ as the Ricci flow iteration only provides enough equations to calculate updates of edge weights, thus if $W_i$ depends on $t$ we cannot compute $R_F^t(i,j)$ over each iteration of (5). We select $W_i = \frac{1}{\deg(i)}$ to normalise the sums in (4), which is important when comparing total Forman-Ricci curvature and network entropy (see below).

We further note that:

$$\frac{\partial R_F(i,j)}{\partial \omega_{ij}} = -\frac{1}{2}(\omega_{ij})^{-1/2} \left[ W_i \sum_{k \neq j} (\omega_{ik})^{-1/2} + W_j \sum_{k \neq i} (\omega_{kj})^{-1/2} \right] < 0 \tag{17}$$

implying that as $\omega_{ij}$ decreases, based on our definition of the distance $d(i,j) = \omega_{ij}$, $i$ and $j$ become "closer", and the Forman-Ricci curvature increases, and vice versa (Fig. 1C). This behaviour is as expected from a curvature. Moreover, considering our Ricci flow construction in (5), if $R_F^t(i,j) > \overline{R_F(i,j)}$ then $d_{t+\Delta t}(i,j) = \omega_{ij}^{t+\Delta t}$ will increase, leading to a reduction in $R_F^t(i,j)$ via (17), driving convergence to $\overline{R_F(i,j)}$.

Thus our choice of Ricci flow construction is computationally efficient, facilitates convergence of the flow towards the normaliser and satisfies all our required and desired properties outlined in the results.

## Investigating the correlation between network entropy and total Forman-Ricci curvature on a simple network

We consider the simple $k$-star network displayed in Fig. 2A, consisting of $k+1$ vertices, of which $k$ have a single edge connecting them to a central vertex $i$. We assign each vertex $l \neq j$ a weight $x_l = 1$ and assign vertex $j$ a weight $x_j = \epsilon > 0$.

Our Forman-Ricci curvature is defined on an edge as follows:

$$R_F(i,j) = \deg(i)^{-1} + \deg(j)^{-1} - (x_i x_j)^{-1/2}$$
$$\left[ \deg(i)^{-1} \sum_{k \neq j} (a_{ik} x_i x_k)^{1/2} + \deg(j)^{-1} \sum_{k \neq i} a_{ik}(a_{kj} x_k x_j)^{1/2} \right] \tag{18}$$

whence

$$R_F(i,j) = \deg(i)^{-1} \left[ 1 - \sum_{k \in N(i)\setminus j} \sqrt{\frac{x_k}{x_j}} \right] + \deg(j)^{-1} \left[ 1 - \sum_{k \in N(j)\setminus i} \sqrt{\frac{x_k}{x_i}} \right] \tag{19}$$

Which we denote as:

$$R_F(i,j) = r_F(i|j) + r_F(j|i) \tag{20}$$

for notational ease, where:

$$r_F(i|j) = \deg(i)^{-1} \left[ 1 - \sum_{k \in N(i)\setminus j} \sqrt{\frac{x_k}{x_j}} \right] \tag{21}$$

We note that via (1):

$$\frac{x_k}{x_i} = \frac{x_k / \sum_{l \in N(j)} x_l}{x_i / \sum_{l \in N(j)} x_l} = \frac{p_{jk}}{p_{ji}} \tag{22}$$

which gives us the alternative expression, which can be helpful when considering stochastic matrices

$$r_F(i|j) = \deg(i)^{-1}\left[1 - \sum_{k\in N(i)\setminus j}\sqrt{\frac{p_{ik}}{p_{ij}}}\right]. \qquad (23)$$

Employing the results above it is a simple deduction that for our toy network:

$$\begin{cases} p_{ij} = \frac{\epsilon}{k+\epsilon-1} & \\ p_{il} = \frac{1}{k+\epsilon-1} & l\neq j \\ p_{li} = 1 & l\neq i \\ p_{lj} = 0 & l\neq i \end{cases}. \qquad (24)$$

The stationary distribution of the network is also easily calculated from (11) as:

$$\begin{cases} \pi_i = \frac{1}{2} & \\ \pi_l = \frac{1}{2(k+\epsilon-1)} & l\neq j,i \\ \pi_j = \frac{\epsilon}{2(k+\epsilon-1)} & \end{cases}. \qquad (25)$$

It is also clear that the local entropies will satisfy:

$$\begin{cases} S_i = -\frac{\epsilon}{k+\epsilon-1}\log\left(\frac{\epsilon}{k+\epsilon-1}\right) - \frac{k-1}{k+\epsilon-1}\log\left(\frac{1}{k+\epsilon-1}\right) \\ S_l = 0 \quad l\neq i \end{cases}. \qquad (26)$$

The network entropy of this network is thus simply:

$$S_R = -\frac{1}{2}\left[\frac{\epsilon}{k+\epsilon-1}\log\left(\frac{\epsilon}{k+\epsilon-1}\right) + \frac{k-1}{k+\epsilon-1}\log\left(\frac{1}{k+\epsilon-1}\right)\right] \qquad (27)$$

Which is a convex function of $\epsilon$ maximal at $\epsilon = 1$ (Fig. 2B).

We now consider the total Forman-Ricci curvature, defined by:

$$R_F = \sum_{l\in V}\pi_l R_F(l) \qquad (28)$$

where

$$R_F(l) = \frac{1}{\deg(l)}\sum_{r\in V}a_{lr}R_F(l,r). \qquad (29)$$

In our example, the following can be deduced from equation (23):

$$\begin{cases} r_F(l|i) = 1 & l\neq i \\ r_F(i|l) = \frac{1}{k}(3 - k - \sqrt{\epsilon}) & l\neq j \\ r_F(i|j) = \frac{1}{k}\left(1 - (k-1)\sqrt{\frac{1}{\epsilon}}\right) \end{cases}. \qquad (30)$$

Which allows the calculation of

$$\begin{cases} R_F(i) = \frac{1}{k}\left[k + \frac{1}{k}\left(1 - (k-1)\sqrt{\frac{1}{\epsilon}}\right) + \frac{k-1}{k}(3-k-\sqrt{\epsilon})\right] \\ R_F(j) = 1 + \frac{1}{k}\left(1 - (k-1)\sqrt{\frac{1}{\epsilon}}\right) \\ R_F(l) = 1 + \frac{k-1}{k}(3-k-\sqrt{\epsilon}) \quad l\neq j \end{cases}. \qquad (31)$$

Whence

$$R_F = \frac{1}{2k}\left[k + \frac{1}{k}\left(1 - (k-1)\sqrt{\frac{1}{\epsilon}}\right) + \frac{k-1}{k}(3-k-\sqrt{\epsilon})\right]$$
$$+ \frac{\epsilon}{2(k+\epsilon-1)}\left[1 + \frac{1}{k}\left(1 - (k-1)\sqrt{\frac{1}{\epsilon}}\right)\right] \qquad (32)$$
$$+ \frac{k-1}{2(k+\epsilon-1)}\left[1 + \frac{k-1}{k}(3-k-\sqrt{\epsilon})\right].$$

**Network entropy and total Forman-Ricci curvature comparison**

Network entropy was calculated on each gene expression sample as described above. Forman-Ricci curvature was computed over an edge $(i,j)$ using the following expression:

$$R_F(i,j) = \deg(i)^{-1} + \deg(j)^{-1} - (x_i x_j)^{-1/2}$$
$$\left[\deg(i)^{-1}\sum_{k\neq j}(a_{ik}x_i x_k)^{1/2} + \deg(j)^{-1}\sum_{k\neq i}a_{ik}(a_{kj}x_k x_j)^{1/2}\right] \qquad (33)$$

Nodal average Forman-Ricci curvature was computed as previously described[22,25] via:

$$Ric_i(\mathbf{x}) = \frac{1}{\deg(i)}\sum_{j\in V}a_{ij}R_F(i,j) \qquad (34)$$

and network average, or total Forman-Ricci curvature was computed via:

$$Ric(\mathbf{x}) = \sum_{i\in V}\pi_i(\mathbf{x})Ric_i(\mathbf{x}), \qquad (35)$$

where $(\pi_i(\mathbf{x}))_{i=1}^n$ is the stationary distribution of $P(\mathbf{x})$.

The choice of node weights for our Forman-Ricci curvature $W_i = \frac{1}{\deg(i)}$ is important here as it ensures that the upper bounds of each of the two sums comprising edge-wise Forman-Ricci curvature defined in (4) are not dependent on node degree, and so nodal average Forman-Ricci curvature is also independent of degree. This is required as the local entropy of a node $i$ (defined in (8)) takes values on $[0, \deg(i)]$ and thus has a degree dependence. We define total Forman-Ricci curvature here, to mirror network entropy, as a weighted sum of nodal average curvatures, using the stationary distribution $(\pi_i(\mathbf{x}))_{i=1}^n$ as the weights. Our choice of node weights $W_i = \frac{1}{\deg(i)}$ thus prevents total Forman-Ricci curvature and network entropy from correlating purely because of a shared degree dependence. We note that while our choice of $W_i$ prevents degree dependence of edge wise and nodal Forman-Ricci curvature, the use of the stationary distribution in calculation of total Forman-Ricci curvature introduces the relative biological importance of hub nodes[9].

Associations between network entropy and total Forman-Ricci curvature were assessed using Pearson correlation with significance at the 5% level.

**Computing linear and Ricci flow trajectories between time-ordered gene expression samples**

Trajectories for time course gene expression data were derived via two approaches, a null Euclidean straight line trajectory and by employing our discrete normalised Ricci flow. For both approaches the first gene expression time point ($\mathbf{x}^0$) was used as a starting state and the final time point ($\mathbf{x}^T$) was the end state. Intermediate time points were not used in the derivation of the trajectory only for its validation.

For normalised discrete Ricci flow we employ the following expression described above:

$$d_{t+\Delta t}(i,j) = d_t(i,j) + \Delta t(Ric(\mathbf{x^t})_{(i,j)} - \overline{Ric}_{(i,j)})d_t(i,j). \tag{36}$$

This flow will deform the weight on an edge of the PIN at a rate proportional to the difference between the edge curvature at a starting state and a final state determined by the normaliser.

We set the normaliser of our Ricci flow as the Forman-Ricci curvature calculated at the final time point $T$: $\overline{Ric}_{(i,j)} = R_F^T(i,j)$. The time increment $\Delta t$ was selected empirically. If $\Delta t$ is too large then negative values of the incremented distance $d_{t+\Delta t}$ are possible, which are not acceptable by definition, however, if $\Delta t$ is very small convergence of the Ricci flow to the normaliser will require a great number of iterations and will not be computationally practical. We therefore considered a range of values for $\Delta t \in \{10^{-3}, ..., 10^{-1}\}$. For each gene expression time course, we implemented one time step of the Ricci flow from the first time point $\mathbf{x^0}$ using each $\Delta t$ value and selected the optimal $\Delta t$ as the largest which does not admit negative values of $d_{0+\Delta t}$. For both time courses considered this value was $\Delta t = 0.06$.

We note that the maximal value of $\Delta t$ which does not admit negative values of $d_{0+\Delta t}$ can also be derived theoretically and depends on the differences between the edge-wise Forman-Ricci curvatures at $t = 0$ and those of the normaliser via:

$$\Delta t^* = \min_{(i,j) \in E^*} \left( \frac{1}{(\overline{Ric}_{(i,j)} - Ric(x^0)_{(i,j)})} \right), \tag{37}$$

where, $E^* = \{(i,j) \in E : \overline{Ric}_{(i,j)} - Ric(x^0)_{(i,j)} > 0\}$. For both time courses considered $\Delta t^* \in [0.06, 0.065]$ and Ricci flow was thus implemented using close to the maximal value of $\Delta t$ possible. Smaller values of $\Delta t$ can be used to obtain a more fine-grain approximation of the network rewiring trajectory, at the cost of increased computation time and the need for more iterations before convergence.

For both gene expression time courses, we found that after 150 iterations the normalised Ricci flow converged very close to the normaliser, with little change in $d_{t+\Delta t}$ with subsequent iterations, we thus selected 150 as the optimal number of iterations in the flow. We note that by construction the final transcriptomic time point will always be closest to the end of the trajectory. As the number of iterations is selected as sufficiently large to ensure convergence, rather than the minimum number of iterations required for convergence, the end of the trajectory represents signalling in a steady state, as opposed to the precise moment gene expression matches the final time point.

To derive the Euclidean linear trajectory null model, from the starting gene expression time point to the final, we constructed a straight line from $W^0 = (a_{ij}x_i^0x_j^0)_{i,j \in V}$ to $W^T = (a_{ij}x_i^Tx_j^T)_{i,j \in V}$ in $\mathbb{R}^{n \times n}$. We selected 150 equally spaced points along this line via the following expression

$$W^t(i,j) = W^0(i,j) + \frac{t(W^T(i,j) - W^0(i,j))}{150}. \tag{38}$$

## Comparing inferred trajectories to true time course gene expression data

For both normalised discrete Ricci flow and the Euclidean linear trajectory null model we derived a trajectory described by 150 discrete points from the starting gene expression state to the final, as above. Each of these discrete data points can be transformed into a prediction of the weighted network: $W_p(\mathbf{x^r}) = a_{ij}x_i^rx_j^r$ for $r \in \{1, ..., 150\}$. In the case of the Euclidean trajectory, the inferred point is exactly this weighted network, while for the normalised Ricci flow $W_p(\mathbf{x^r}) = (1/d_r(i,j))_{i,j \in V}$.

For each true intermediate time point in the gene expression time course $\{1, ..., T-1\}$ we computed the Euclidean distance between each

of the 150 predictions of $W_p(\mathbf{x^r})$ in each inferred trajectory and the true data points $\{W(\mathbf{x^1}), ..., W(\mathbf{x^{T-1}})\}$.

The value of $r$ which minimised the distance between $W_p(\mathbf{x^r})$ and $W(\mathbf{x^t})$ was considered the point along the trajectory which most closely corresponded to the true gene expression trajectory at time $t$.

The association between the trajectory points corresponding to the measured time points and the true intermediate time points themselves (excluding starting and ending time points) was assessed via Pearson correlation, with significance at the 5% level.

## Entropy and Ricci curvature on networks and metric-measure spaces

A connection between Ricci curvature and relative entropy has been explored in the setting of metric-measure spaces by several investigators[24,40,48]. Formally let $(M, d, m)$ be a metric-measure space, where $(M, d)$ is a metric space and $m$ is a measure on the Borel $\sigma$-algebra of $M$, the authors typically aim to define a notion by which $(M, d, m)$ has a Ricci curvature bounded below by $K \in \mathbb{R}$ and explore the consequences. To do so they consider the metric space $P_2(M) = (P(M), W_2)$, associated with the metric space $(M, d)$, where $P(M)$ is the space of Borel probability measures on $M$ and $W_2$ is the Wasserstein-2 distance. $W_2$ is a distance measure commonly used in optimal transport, to provide intuition if $m_1, m_2 \in P(M)$ then $W_2(m_1, m_2)^2$ is the smallest cost of transporting the total mass from the measure $m_1$ to the measure $m_2$, where the cost of transporting a unit mass between points $a_1$ and $a_2 \in M$ is $d(a_1, a_2)^2$. Employing results on displacement convexity along geodesics in $P(M)$, a connection between an entropy functional defined on $P(M)$ and the Ricci curvature of $(M, d, m)$ can be proposed.

Formally, using the notation of Strum 2006[40], we define a relative entropy functional with respect to $m$ on $P(M)$ via:

$$\text{Ent}(\nu|m) = \int_M \frac{d\nu}{dm} \log\left(\frac{d\nu}{dm}\right) dm \tag{39}$$

It has been proposed (based on results for Riemannian manifolds[48]) that $(M, d, m)$ has Ricci curvature bounded below by $K \in \mathbb{R}$ if and only if, for any $\nu_0, \nu_1 \in P(M)$, where $\text{Ent}(\nu_0|m), \text{Ent}(\nu_1|m) < \infty$, there exists a geodesic $\gamma: [0, 1] \to P(M)$, where $\gamma(0) = \nu_0$ and $\gamma(1) = \nu_1$ such that:

$$\text{Ent}(\gamma(t)|m) \leq (1-t)\text{Ent}(\gamma(0)|m) + t\text{Ent}(\gamma(1)|m) \\ - \frac{K}{2}t(1-t)W_2(\gamma(0), \gamma(1))^2. \tag{40}$$

Sandhu et al.[19], use this statement to infer a positive correlation between an entropy defined as the negative of $\text{Ent}(\cdot|m)$ and the Ricci curvature of $(M, d, m)$.

In our setting of networks, there is not an unambiguous way to map to a metric-measure space. The definition of the space $(M, d, m)$ could have many choices in terms of network topology as well as vertex and edge weights. Moreover, the definition of Forman-Ricci curvature applied to networks is again non-unique, depending on edge and vertex weights and the validity of this curvature depends upon an interpretation of the network as a cell complex approximation to a Riemannian manifold. The definition of network entropy as an entropy rate is also not equivalent to the definition of $\text{Ent}(\cdot|m)$, and again the choice of $m$ for the network setting is non-unique. Collectively this highlights a distinction between the network setting and metric-measure spaces, and results in one setting cannot be expected to be valid in the other, in particular correlation between entropy and curvature.

## Statistics and reproducibility

The association between network entropy and total Forman-Ricci curvature in transcriptomic data sets was evaluated using Pearson's

correlation coefficient. The comparison between network entropy and total Forman-Ricci curvature in cancerous and healthy single cells was evaluated using two-tailed Wilcoxon tests. The association between closest pass Ricci flow iteration/straight line trajectory iteration and true differentiation time was evaluated using Pearson's correlation coefficient. No statistical method was used to predetermine the sample size. No data were excluded from the analyses. The experiments were not randomised. The Investigators were not blinded to allocation during experiments and outcome assessment.

### Reporting summary

Further information on research design is available in the Nature Portfolio Reporting Summary linked to this article.

## Data availability

All relevant data supporting the key findings of this study are available within the article. The Normalised read count data corresponding to RNA-sequencing used in this study are available in the GEO database[49] under the following accession codes. The data describing scRNAseq of 1018 single cells assayed at different stages of multipotency and alongside data describing 758 single cells assayed at 6 distinct time points during ESC differentiation[42] are available in the GEO database under accession code GSE75748. The data describing scRNAseq of 1257 malignant and 3256 healthy single cells from 19 patients with malignant melanoma[43] are available in the GEO database under accession code GSE72056. The data describing scRNAseq of 272 malignant and 160 healthy cells from patients with colorectal cancer[44] are available in the GEO database under accession code GSE81861. Our data set describing healthy myoblast differentiation at 8 distinct time points[45] is available in the GEO database under accession codes GSE102812 and GSE123468. Source data are provided in this paper.

## Code availability

The R code developed for the analysis presented in the paper is accessible in the following Github: https://github.com/anthbapt/Cellular-differentiation-trajectories-with-Ricci-flow and the used version of the code is deposited in Zenodo with https://doi.org/10.5281/zenodo.10469562[50].

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

## Acknowledgements

All authors gratefully acknowledge funding from the Turing-Roche Strategic Partnership, and Prof. Ginestra Bianconi for interesting discussions.

## Author contributions

C.R.S.B. and B.D.M. designed research; A.B. and C.R.S.B. performed research; A.B. and C.R.S.B. analysed data; C.R.S.B. created numerical code, with contributions from A.B.; A.B., B.D.M., and C.R.S.B. wrote the paper.

## Competing interests

The authors declare no competing interests.
