## [Peer Review File · Nature Communications]

Charting cellular differentiation trajectories with Ricci flowEditorial Note: This manuscript has been previously reviewed at another journal that is not operating a transparent peer review scheme. This document only contains reviewer comments and rebuttal letters for versions considered at *Nature Communications*.

Reviewer #1 (Remarks to the Author):

This manuscript investigates the connection between the entropy and Ricci curvature in the context of cellular processes, such as cellular differentiation. The authors state that predicting dynamic biological network rewiring remains an open problem. The authors apply the Ricci curvature and Ricci flow to biological network rewiring. The authors investigate the relationship between network entropy and Forman-Ricci curvature, theoretically and empirically on single-cell RNA-sequencing data, and demonstrate that the two measures do not always positively correlate, as it was previously suggested. The authors employ Ricci flow to derive network rewiring trajectories from stem cells to differentiated cells, accurately predicting true intermediate time points in gene expression time courses. I find the manuscript quite interesting and worthy of publication, but a couple of issues require further attention:

A) [p6, §2.1.1] "From this association it was concluded that the negative of the relative entropy and Ricci curvature are positively correlated [19]. We note, however, that the network setting is not equivalent to metric-measure spaces. In particular network entropy (an entropy rate) is not equivalent to the relative entropy described by [24]."

Since this is an important statement supporting the paper's main claim as to why the two metrics do not always correlate, more elaboration about this is needed.

B) [p.8, §2.2] "To gain intuition we investigated the association between our version of Forman-Ricci curvature..."

Is this based on the chosen definition for the node weights = $1/\text{deg}(i)$? Or are there other modifications to the F-R curvature formulation used in this section? It is not clear from the text.

C) [p.12, Fig. 3c] While the overall trend seems negatively correlated for the melanoma patients data, the F-R curvature is positively correlated for network entropy > 0.9 . Additionally, for the melanoma cells (red), Pearson's $r = -0.009$ is reported, which indicates no linear relationship. Is there any biological explanation for the piece-wise trend?

D) [Discussion] "By investigating our framework in a simple analytically tractable setting, we prove that network entropy and our total Forman-Ricci curvature are not guaranteed to be positively correlated."

Are there any formal proofs to indicate as such? This could be related to Comment #1.

F) A recent related paper explores the phase transition detection in time-varying complex networks using the Forman-Ricci curvature. "A unified approach of detecting phase transition in time-varying complex networks." Scientific Reports 13, no. 1 (2023): 17948.
<https://www.nature.com/articles/s41598-023-44791-3>

Reviewer #2 (Remarks to the Author):

I am satisfied with the update and response.

Response to Reviewers

Reviewer #1 (Remarks to the Author):

This manuscript investigates the connection between the entropy and Ricci curvature in the context of cellular processes, such as cellular differentiation. The authors state that predicting dynamic biological network rewiring remains an open problem. The authors apply the Ricci curvature and Ricci flow to biological network rewiring. The authors investigate the relationship between network entropy and Forman-Ricci curvature, theoretically and empirically on single-cell RNA-sequencing data, and demonstrate that the two measures do not always positively correlate, as it was previously suggested. The authors employ Ricci flow to derive network rewiring trajectories from stem cells to differentiated cells, accurately predicting true intermediate time points in gene expression time courses. I find the manuscript quite interesting and worthy of publication, but a couple of issues require further attention:

We thank the reviewer for their positive and accurate summary of our results and for stating that our work is worthy of publication in *Nature Communications*.

A) [p6, §2.1.1] “From this association it was concluded that the negative of the relative entropy and Ricci curvature are positively correlated [19]. We note, however, that the network setting is not equivalent to metric-measure spaces. In particular network entropy (an entropy rate) is not equivalent to the relative entropy described by [24].”

Since this is an important statement supporting the paper's main claim as to why the two metrics do not always correlate, more elaboration about this is needed.

We appreciate this point, and now include an additional more technical section in the Materials and Methods (p.25-26), referenced when these measures are first introduced on p.6, where the reviewer indicates. In this technical section we briefly outline the results of Lott and Villani, 2009 and Sturm 2006, on metric measure spaces, from which Sanhu et al., 2015 deduced a correlation between a relative entropy and Ricci curvature. We highlight that the measures of curvature and entropy employed in the metric measure space setting are not equivalent to those used in the network setting, so theoretical results cannot be directly translated from metric measure spaces to networks.

B) [p.8, §2.2] “To gain intuition we investigated the association between our version of Forman-Ricci curvature...”

Is this based on the chosen definition for the node weights = $1/\text{deg}(i)$? Or are there other modifications to the F-R curvature formulation used in this section? It is not clear from the text.

We appreciate this point raised by the reviewer. Throughout the manuscript we only use one definition of edge-wise Forman-Ricci curvature (FRC), we never change the definition of edge or node weights. Our edge-wise FRC, $Ric(i, j)$, is defined using node weights $W_i = 1/\text{deg}(i)$ and edge weights $\omega_{ij} = a_{ij}/x_i x_j$, the full expression is provided in equation (6) on p.8 at the end of section 2.1. To compare this measure with network entropy (a network level measure) in sections 2.2 and 2.3, we average edge-wise FRC at the network level, using the established convention of the field (e.g., Sandhu et al., 2015, Murgas et al., 2022) defined in equation (2) on p.6 Section 2.1.1.

We now refer the reader to these two equations on p.8 section 2.2 (where the reviewer indicates), so that the construction of the FRC used is clear.

C) [p.12, Fig. 3c] While the overall trend seems negatively correlated for the melanoma patients data, the F-R curvature is positively correlated for network entropy > 0.9 . Additionally, for the melanoma cells (red), Pearson's $r = -0.009$ is reported, which indicates no linear relationship. Is there any biological explanation for the piece-wise trend?

The reviewer raises the important point that there are two regimes of association between network entropy and total FRC in the biological data. From our theoretical investigation of the k-star network we expected this. In the k-star we found the measures are positively correlated for 'many for one' signalling, where most information sent from the central node is distributed across a wide number of interaction partners. Conversely, negative correlation occurs in 'one for many' signalling, where the central node preferentially signals to one interaction partner. In the case of our simple k-star network we analytically demonstrate the turning point between these regimes, and its dependence on the two parameters k and ϵ in Fig 2D.

As we comment in the manuscript using biological intuition we suggest 'many to one' signalling is more likely in stem cells, which must maintain the option to perform a variety of functions. We confirm this intuition by observing a positive correlation between the network measures in stem cells (Fig 3A-B). Conversely, 'one to many' signalling is intuitively more likely in differentiated cells which must perform a specific function and not deviate. We again confirm this intuition by observing a negative correlation between the network measures on healthy differentiated cells (blue points, Fig 3C-D).

In the case of malignant cells, we see a less negative correlation or even no correlation in our measures, as the reviewer remarks. One of the hallmarks of malignant cells is anaplasia – the observation that cancerous cells appear less differentiated than their healthy counterparts. Anaplasia is typically quantified by histological grade, where tumour cells are compared morphologically to their healthy counterparts and assigned a low grade if they appear similar, or a high grade if they have lost the appearance associated with specialised function (e.g., loss of villi on bowel cancer cells). Anaplastic malignant cells gain some of the hallmarks of stem cells, such as a higher proliferative capacity, they also gain additional functions, including those which facilitate metastasis.

Extending our biological intuition, we suggest that the loss of negative correlation between our measures on malignant cells, may represent an increase in 'many to one' signalling compared to healthy cells, which would be expected under anaplastic transformation of healthy cells to malignant cells. Highly anaplastic cells may attain the signalling regime more characteristic of stem cells (i.e., a positive correlation between network entropy and total FRC).

We now comment on this interpretation in the updated discussion on p.16.

D) [Discussion] "By investigating our framework in a simple analytically tractable setting, we prove that network entropy and our total Forman-Ricci curvature are not guaranteed to be positively correlated."

Are there any formal proofs to indicate as such? This could be related to Comment #1.

We thank the reviewer for this comment. The results of section 2.2 encompass a formal disproof by counterexample to the claim: 'Total Forman-Ricci curvature and network entropy are guaranteed to be positively correlated'. In our response to comment 1 of the reviewer we have highlighted a disconnect between the metric measure space setting and the network setting, hence our disproof of the above claim in the network setting does not have any implications for the formal proofs in the

metric measure space setting. We highlight this point in the updated Materials and Methods on p.25-26.

F) A recent related paper explores the phase transition detection in time-varying complex networks using the Forman-Ricci curvature. "A unified approach of detecting phase transition in time-varying complex networks." Scientific Reports 13, no. 1 (2023): 17948.

<https://www.nature.com/articles/s41598-023-44791-3>

We thank the reviewer for highlighting this interesting and highly relevant paper which we now include in the introduction on p.3.

Reviewer #2 (Remarks to the Author):

I am satisfied with the update and response.

We thank the reviewer for their time taken reviewing our updated manuscript and are delighted that they are satisfied the paper is now suitable for publication.

Reviewer #1 (Remarks to the Author):

I have now read the revised manuscript and the responses provided by the authors and I remained at my positive impression about this manuscript and work. I think the authors have address all issues raised, so I recommend acceptance of this manuscript in its current form without alterations. This is a very interesting study that will stimulate other ideas in the community especially in relation to the concept of phase transitions.